# Parkinson’s Disease and the Metal–Microbiome–Gut–Brain Axis: A Systems Toxicology Approach

**DOI:** 10.3390/antiox11010071

**Published:** 2021-12-28

**Authors:** Lady Johanna Forero-Rodríguez, Jonathan Josephs-Spaulding, Stefano Flor, Andrés Pinzón, Christoph Kaleta

**Affiliations:** 1Research Group Bioinformatics and Systems Biology, Instituto de Genetica, Universidad Nacional de Colombia, Bogotá 111321, Colombia; ljforeror@unal.edu.co (L.J.F.-R.); ampinzonv@unal.edu.co (A.P.); 2Research Group Medical Systems Biology, Christian-Albrechts-Universität Kiel, Brunswiker Straße 10, 24105 Kiel, Germany; s.flor@iem.uni-kiel.de (S.F.); c.kaleta@iem.uni-kiel.de (C.K.)

**Keywords:** heavy metals, neurotoxicity, Parkinson’s disease, ROS, human microbiome, systems toxicology

## Abstract

Parkinson’s Disease (PD) is a neurodegenerative disease, leading to motor and non-motor complications. Autonomic alterations, including gastrointestinal symptoms, precede motor defects and act as early warning signs. Chronic exposure to dietary, environmental heavy metals impacts the gastrointestinal system and host-associated microbiome, eventually affecting the central nervous system. The correlation between dysbiosis and PD suggests a functional and bidirectional communication between the gut and the brain. The bioaccumulation of metals promotes stress mechanisms by increasing reactive oxygen species, likely altering the bidirectional gut–brain link. To better understand the differing molecular mechanisms underlying PD, integrative modeling approaches are necessary to connect multifactorial perturbations in this heterogeneous disorder. By exploring the effects of gut microbiota modulation on dietary heavy metal exposure in relation to PD onset, the modification of the host-associated microbiome to mitigate neurological stress may be a future treatment option against neurodegeneration through bioremediation. The progressive movement towards a systems toxicology framework for precision medicine can uncover molecular mechanisms underlying PD onset such as metal regulation and microbial community interactions by developing predictive models to better understand PD etiology to identify options for novel treatments and beyond. Several methodologies recently addressed the complexity of this interaction from different perspectives; however, to date, a comprehensive review of these approaches is still lacking. Therefore, our main aim through this manuscript is to fill this gap in the scientific literature by reviewing recently published papers to address the surrounding questions regarding the underlying molecular mechanisms between metals, microbiota, and the gut–brain-axis, as well as the regulation of this system to prevent neurodegeneration.

## 1. Introduction

### 1.1. What Is Parkinson’s Disease?

Parkinson’s disease (PD) is the second most common neurodegenerative disorder and affects nearly 1% of the population above 60 years of age [1]. PD is characterized by motor and non-motor symptoms, characterized by striatal dopamine depletion and alterations in neurochemicals [2]. Although motor impairment is critical to disease development, non-motor symptoms become evident prior to the emergence of motor dysfunction [3]. PD non-motor symptoms are related to gastrointestinal (GI) dysfunction, followed by motor dysfunction, such as excessive salivation, dysphagia, nausea, and constipation in approximately 30–80% of patients [4,5]. The direct cause of PD onset is unknown and is loosely related to genetic background (5% to 10%); environmental factors in idiopathic PD are postulated to trigger pathogenesis [6,7,8].

### 1.2. Gut–Brain Axis: Braak’s Hypothesis

The GI microbiome is a dynamic ecosystem comprising microbial communities impacted by individual diet, host-derived metabolites, and environmental exposures. More so, the human microbiome is essential in bidirectional gut–brain communication; brain and intestinal development, function, and homeostasis are likely mediated by the gut microbiome [9,10,11,12,13]. Braak and colleagues developed a hypothetical framework to explain the influence of environmental factors on PD development [14]. For example, invasive microbiota from the gut during dysbiosis may induce a pro-inflammatory environment, thereby promoting PD onset [15,16]. Pathological microbial agents retrogradely promote α-synuclein formation and spread into the brainstem, locus coeruleus complex, substantia nigra, and the cerebral cortex [17,18]. However, some studies suggest that the presence of α-synuclein within the enteric nerves of the gut tissues should not be used to infer PD pathology [19]. Overall, in PD patients, microbiota–gut–brain axis communication is likely impaired, whereas GI dysfunction symptoms are observed in over 80% of PD subjects [5,20]. However, general research is necessary to elucidate communication mechanisms between gut microbiota in relation to the gut–brain axis [12].

With regard to recent observations, there are wider theories to inform our understanding of the role of the intestinal environment as a significant influencer in the gut–brain axis theory. Presently, it is well-described that irritable bowel disease (IBD) and other inflammatory diseases of the GI are closely associated with PD risk [21,22,23,24]. For example, within a Danish IBD cohort (*n* = 76,477), patients had an 22% increased risk for PD; from a USA IBD cohort (*n* = 144,018) an increased risk of 28%; patients within a Swedish population-based study (*n* = 39,000) had an incidence rate of 30%; and a Taiwanese nationwide cohort study (*n* = 8375) suggested that patients with IBD had a 43% increased incidence of PD onset [21,22,23,24]. However, treatment with anti-tumor necrosis factor therapies (anti-TNF) was demonstrated to substantially mitigate the incidence of PD by reducing systemic inflammation, which drives the tandem pathogenesis of IBD and PD, demonstrating a causal link of the gut–brain axis [22]. Specifically, anti-TNF was shown to reduce the incidence of PD by 78%, as compared to individuals who do not take the therapy [22].

Microbiota are also essential for provoking PD or preventing the further progression of disease. For example, in a pilot study to investigate pathogenic growth in relation to PD, it was found that patients with PD had a 10-times-increased relative risk of being colonized by *Heliobacter pylori* [25]. In a subsequent study investigating the role of eradicating *H. pylori* from PD patients, with a positive C-urea breath test used to identify *H. pylori*, it was found that eliminating this pathogenic bacterium improved host response to levodopa (L-DOPA) and numerous patients’ symptoms (motor movement, non-motor functions, and overall quality of life) that are associated with PD onset [26]. In contrast, evidence exists describing some beneficial bacteria in patients with PD. Comparing metagenomic shotgun sequencing of the gut microbiome between 76 PD patients and 78 healthy controls, the authors identified *Akkermansia muciniphila* as enriched in PD patients after controlling for medication that was provided to intervene against PD [27]. Generally, this microbe is known to confer several benefits to humans such as inciting longer gastrointestinal passage times [28,29], improving the GI barrier functioning [30], and improving anti-inflammatory status [31].

To date, there are a lack of studies investigating the link between metal-induced oxidative stress upon the gut microbiota and neurodegeneration, independently and as a system [16]. Heavy metals promote oxidative stress, thereby altering gut barrier permeability and, subsequently, inflammation, leading to amplified heavy metal absorption and traffic into the brain [32]. The gut microbiota also affects intestinal metal absorption by modulating the barrier function and bioremediation of heavy metals in the gut [33,34]. Therefore, interventions to mitigate dietary toxic metals likely reduce the inflammatory burden on beneficial gut microbiota and potentially PD onset. For example, numerous probiotic strains from the genera *Lactobacillus* and *Bifidobacterium* have known bioremediation effects on heavy metals [35,36]. To better decipher the systems-based link between the metal–microbiome–gut–brain axis, it is important to interpret how microbial dysbiosis, in relation to oxidative stress from metals, leads to a pro-inflammatory environment and how the gut microbiome influences the remediation of xenobiotic metals [37]. Therefore, this review will summarize the current understanding of the metal–microbiome–gut–brain axis and propose future perspectives to mitigate heavy metal toxicity through gut-associated microbiome chelation, specifically, the process in which microbiota remove toxic metals from the body by producing molecules, which reduce toxicity and oxidative stress prior to neurodegenerative disorders [38]. Overall, the presented cumulative review is an attempted combination of recent evidence from the two fields of PD and the metal microbiome to identify both overlaps and gaps; our aim is to provide suggestions through the lens of systems toxicology to merge both fields.

### 1.3. Neurotoxicity and Parkinson’s Disease

Biological organisms require 13 metals, 9 of which are essential trace metals for creating organic building blocks and regulating homeostatic processes such as catalyzing enzymes for basic metabolic or biochemical processes. Approximately one-third of biological proteins and 40% of enzymes rely on metal ions to function [39]. The brain generally contains the greatest concentrations of metals in the human body, including Fe, Cu, Zn, and Mn [40]. For example, both Zn and Mn are found in the midbrain and are essential for neuronal excitement, synaptic transmission, the creation of new myelin, and regulation of oxidative stress [41,42]. Conversely, when these metals are above normal homeostatic balance, they are associated with PD [43]. Specifically, increased occupational exposure to Fe or Mg was found to double the risk of developing PD [44]. Improper metal binding associated with oxidative stress leads to protein misfolding and aggregation [45]. Metals with a high affinity to sulfhydryl groups, alter dopamine neurotransmission, reduce expression of D2 dopamine receptor sites, and impair proteins essential for maintaining cellular homeostasis [46]. Protein misfolding is the hallmark of numerous neurodegenerative diseases such as Alzheimer’s Disease, Lewy Body Dementia, Huntington’s Disease, and prion diseases, and is similarly underlined with metal stress [47]. The association between metal exposure, the accumulation of misfolded proteins (especially α-synuclein) found in Lewy bodies and PD is not well understood at the mechanistic level and is likely associated with an increase in ROS [45]. Regardless, the potential to control metal exposure and maintain cellular homeostasis in PD may prevent improper neuroprotein folding prior to apoptosis. There is a need to review the literature surrounding the topic of metabolic outcomes directed by heavy metals exposure and oxidative stress prior to neuropathology.

#### 1.3.1. Manganese (Mn)

Mn is a trace metal found throughout various tissues and is essential for enzymatic processes, the biosynthesis of amino acids, protein formation, and carbohydrate metabolism [47,48]. Environmental exposure to Mn primarily occurs by ingesting leafy vegetables, grains, and nuts which enter the circulatory system through passive diffusion [49]. Chronic exposure to Mn is positively correlated with the hallmark characteristics of PD pathology, such as dopaminergic nerve deterioration and impairment of the basal ganglia, which includes the globus pallidus, substantia nigra, subthalamic nucleus and striatum [50]. Different valence states or species of Mn (Mn^2+^ or Mn^3+^) play significant roles in Mn neurotoxicity, in which Mn^3+^ is more toxic than Mn^2+^ [51]. Typically, Mn is found in portal circulation, 80% of Mn^2+^ is found to be associated with α-macroglobulin or albumin. Toxic Mn^3+^ species account for 1% of the total Mn found in the body; Mn^3+^ associates with transferrin endosomes during tissue circulation and Mn^2+^ diffuses through tissues, including the kidneys, bones, liver, and the brain [52].

Within the brain, Mn^3+^ accumulates and enters neurons by binding to transferrin endosomes and localizes within brain tissue [48,53]. Specifically, Divalent Metal Transporter 1 (DMT1) is highly expressed in brain tissue, especially in dopamine-rich areas of the basal ganglia and cortex, which explains Mn association with the brain [54,55]. Mn exposure affects the transcription and translation of α-synuclein and through the activation of MAPK, thus beginning the apoptosis cascade signaling, and results in dopaminergic neuron apoptosis [56,57]. Mn stress leads to the dysregulation of key homeostatic functions, which are essential for degrading proteins (such as autophagy, proteasomes, and endosomal trafficking) [58]. Chronic metabolic dysequilibrium linked to Mn initiates conformational changes leading to the aggregation of α-synuclein, which can become harmful to neuronal cells [59]. This suggests that Mn and other metal ions incite the production of ROS, leading to conformation changes and the aggregation of α-synuclein [53,58]. However, recent observations of human α-synuclein have identified a neuroprotective role in the mitigation of dopaminergic degeneration linked to acute Mn exposures [60]. Conversely, neuroprotective effects are lessened when neurons are chronically exposed to Mn, leading to accelerated oxidation and misfolding of α-synuclein [60].

Interestingly, Mn-induced Parkinsonism is not correlated with a significant loss of dopaminergic neurons in non-human primates [61,62]. This highlights that Mn exposure can be regulated at the environmental level to prevent PD outcomes. While occupational exposure to Mn was significantly mitigated through public health interventions and regulations, there is still a great need for the differentiation between chronic Mn neurotoxicity and PD symptoms [63]. Mn neurotoxicity is regularly confused with idiopathic PD symptoms; while both differ in several ways, they also share several similarities. Generally, the distribution of Mn within the brain elicits symptoms that are similar to PD; however, clinical observations of these findings are not as common as a PD diagnosis [63]. For example, Mn impacts the neurons of both the globus pallidus striatum and substantia nigra pars reticulata, but accumulates in a lesser proportion in the substantia nigra pars compacta [64]. This occurs through the induction of alterations in energy deficits, protein aggregation, ubiquitin-proteasome system dysfunction, and altered mitochondria function, all of which are closely related to oxidative stress [45]. Thus, the neurotoxic effects on the globus pallidus, the substantia nigra pars reticulata, and the striatum, which impact the dopaminergic neurons, are typical hallmarks of PD [65,66]. The dysregulation of dopamine transmission from the substantia nigra to the striatum, a region associated with motor symptoms in Mn neurotoxicity, is known to prevent dopamine release, thereby leading to elicit behavioral responses that are similar to both PD and Mn neurotoxicity [64,66]. Additionally, Mn toxicity impairs cells that normally produce the neurotransmitter γ-aminobutyric acid (GABA); this dysfunction is associated with numerous psychiatric disorders such as major depressive disorder, bipolar disorder, and schizophrenia [67,68]. Therefore, there is a need to understand Mn exposure in relation to PD onset in terms of molecular mechanisms. For example, recently, a transcriptomic study investigated expression signatures of human neuroblastoma cells treated with high and low doses of Mn [69,70], revealing perturbed processes involved with nervous system development (axon and cerebral cortex formation). This was manifested as disturbances in the mitochondria cell cycle, ROS production, and chronic inflammatory signaling, which leads to neural cell apoptosis [70].

#### 1.3.2. (Methyl) Mercury (MeHg; Hg)

Hg is a metal that shifts between organic and inorganic phases, thereby inducing differing modes of toxicity. Hg not only bio-accumulates within plants and animals in ecosystems, but is also transported over long distances as a gas following combustion. Hg vapor is stable in a gaseous state and accounts for 90% of Hg within the environment [71]. Inorganic Hg is not physiologically necessary in humans and is toxic in low concentrations. Hg becomes fat-soluble within the tissues of animals after methylation by microbes to become methylmercury (MeHg); this organic state is readily consumed and bioaccumulates from the bottom of food chains. For example, contaminated fish with a high biomass of fatty acids rapidly accumulate MeHg within tissues, as compared to other fish within the same environment. Furthermore, Hg was found to accumulate within the nervous tissue of deceased individuals with neurodegenerative diseases in comparison to healthy controls [72]. Individuals poisoned by chronic MeHg suffer from numerous symptoms linked to central nervous systems (CNS) impairment, such as visual impairments, lethargy, tremors, lack of coordination, and inability to recall memories [73].

Hg differs from other metals with regard to cellular transport; it does not require human macrophages or other immuno-transport mechanisms, rather MeHg is readily able to penetrate the blood–brain barrier (BBB) [74]. Specifically, Hg binds to sulfhydryl groups with a high affinity in comparison to other metals in the human body. When MeHg binds to cysteine groups through molecular mimicry, it is transported through the BBB and directly interacts with glial and neuronal cells [74]. From there, Hg is uptaken through the nerve endings and accumulates in the CNS ganglia where it cannot be detoxified, leading to systemic effects upon the substantia nigra dopaminergic neurons through disruption of tubulin molecules [75,76,77,78]. Dopaminergic neurons consist of axons that are composed of tubulin molecules; however, in the presence of inorganic Hg, this metal binds to the 14-sulfhydryl groups found within the tubulin of the dopaminergic neurons. Through this mechanism, Hg interacts with α-tubulin and β-tubulin, inhibiting the tubulin structural formation of neurons, thus preventing tubulin from binding to Guanosine-5′-triphosphate (GTP) and halting motor neural functions by blocking neurotransmission [79]. Hg further promotes the formation of amyloid-β proteins, the predecessors for amyloid plaques and neurofibrils in the brain; both are associated with the onset of Alzheimer’s Disease and PD [80,81]. Through these pathomechanisms, MeHg exposure leads to apoptosis and neurotoxicity through dysfunctional cell narrowing, chromatin condensation, the modification of cytochrome C flux, and well-described oxidative stress insults to the mitochondria [82].

#### 1.3.3. Iron (Fe)

Fe is a common dietary metal that is essential in humans, assisting in oxygen transport and mitochondrial respiration [83]. Conversely, the over- or underabundance of free Fe in humans is associated with several neurological pathophysiologies [84]. Low Fe, anemia, or low hemoglobin levels throughout the human course of life were correlated with an increased risk of developing PD [85]. Irregular Fe metabolism in humans leads to a lack or excess of Fe in the brain, causing deleterious effects in the PNS or CNS [86]. Incidentally, the argument that Fe accumulation precedes PD and other neurodegenerative diseases is discussed in detail elsewhere [83,87,88].

Dysfunctional Fe metabolism regulates the uptake, sequestration, accumulation, release, and movement of Fe into the substantia nigra pars compacta; these specific pathophysiological mechanisms of Fe metabolism are associated with PD [89]. Specifically, PD patient ferritin, transferrin, and total serum Fe is lower than in healthy controls, illustrating that Fe metabolism and transport is adverse in PD patients [90]. In healthy patients, Fe provides pigmentation and other essential neurophysiological roles; in PD patients, Fe is routinely released from neuromelanin [91]. Approximately 20% of Fe in the substantia nigra is bound to neuromelanin. Impaired Fe metabolism is assessed by the loss of neuromelanin in dopaminergic neurons of PD patients, as observed by magnetic resonance imaging [92,93]. Overall, adverse Fe metabolism highlights the role of metal exposure and dysregulation in individuals, especially in the case of neurodegeneration and idiopathic PD [94].

Fe accumulates within the gray matter of both the basal ganglia and midbrain; this is positively associated with PD onset [95]. Neuronal Fe accumulates in the substantia nigra pars compacta of PD patients, but not in other regions of the brain in postmortem PD patients [96]. High quantities of Fe and a lack of Cu in the substantia nigra pars compacta are typical features of PD onset [31]. When tracking PD patients over 3 years, a longitudinal clinical study identified that Fe increases in the substantia nigra pars compacta and simultaneously decreases in the white matter [97]. A down regulation of DMT1 perturbs Fe metabolism in the brain; this pathology is typically observed in PD brains postmortem. Irregularities in the regulation of this gene lead to chronic ferritin release and neurotoxicity [98].

## 2. Microbiome–Gut–Brain Communication in Health and Disease

The GI tract is a highly interactive surface area between host and environment, as a barrier it is pivotal in regulating immune homeostasis [99]. Six major host cell types comprise the GI barrier: enterocytes, entero-endocrine cells, goblet cells, paneth cells, microfold cells, and stem cells, which form a monolayer barrier that restricts microbial transport and metal ion movement by the expression of transporters [100,101]. Approximately 80% of GI barrier cells are enterocytes; thus, epithelial cells are intermediates between host and luminal interactions in the maintenance of homeostatic functions [100,102,103]. The GI has three major roles as a semipermeable barrier: selective absorption of nutrients, regulation of environmental antigens, and the transport of microorganisms [101,104]. First, the mechanical barrier consists of intestinal epithelial cells and capillary endothelial cells connected with tight or adherens junctions and cadherin proteins, which regulate nutrients [99]. Second, the GI barrier is an immune barrier for controlling environmental antigens. The third, which is extensively covered in this review, is the biological barrier colonized by gut microbiota. However, recent studies highlight that heavy metal exposure alters both the native gut microbiota and intestinal physiology [105,106,107]. Specifically, additional damage due to heavy metals disruption of the gut epithelium leads to a heightened inflammatory response, thus disrupting GI tight junctions and promoting systemic inflammation by inciting changes in microbial abundance and microbial-mediated metabolic changes [108].

The gut–brain axis concept is a fundamental factor for interpreting bidirectional communication and the mechanisms that modulate gut–brain homeostasis, especially during oxidative stress [12]. The central nervous system (CNS), autonomic nervous system (ANS) and enteric nervous system (ENS) influence GI smooth muscle mobility, mucus secretion, and blood flux, thereby modifying intestinal microbiota abundance and, in turn, brain homeostasis and function [12,109]. Microbiome–gut–brain communication is carried out through pathways involving nervous, endocrine, and immune signaling mechanisms [109]. The GI tract is controlled by both intrinsic and extrinsic innervation, in which intrinsic innervation is regulated by the ENS. The ENS is innervated with extrinsic ANS (parasympathetic and sympathetic systems) [110]. The alteration of microbial communities at the gut epithelial barrier modulate both inflammatory responses and metabolic pathways [111,112].

### 2.1. Neuroendocrine and Neuroactive Metabolites Crosstalk

Communication between the brain and gut microbiome is likely modulated through the vagus nerve [113]. The commensal relationship between microbiota and host neurotransmitter synthesis through the enteric nerves is potentially an evolutionary adaptation [114,115]. Various GI microbiota produce metabolites that are identical to the chemical structure of host-derived neuronal metabolites [12]. For example, *Bacillus*, *Escherichia*, and *Proteus* are able to produce the neurotransmitters dopamine and norepinephrine, which are uptaken by the host; molecular signals modulate communication between the microbiota–gut–brain axis and have implications for host physiology [116,117,118,119]. For example, human vagus nerve stimulation is a therapeutic intervention to treat various pathologies, such as refractory depression, IBD, and Crohn’s disease [113,120]. The microbial secondary metabolite indole stimulates the vagus nerve by inducing the expression of c-Fos proteins in the dorsal vagal complex, indicating that vagus nerve activation leads to enhanced anxiety-like behaviors [121]. Within the gut barrier, enteroendocrine cells (EECs) represent 1% of the total epithelial cells and are critical for gut maintenance [12]. Type-L enterochromaffin cells are the most common EECs and directly interact with the intestinal lumen, microbiota, and microbially derived metabolites [122]. EECs produce glucagon-like peptide-1 (GLP-1) and peptide YY (PYY), which stimulate satiation and regulate food intake behaviors [123]. The expansion of these molecules by EECs activate food-derived nutrients, in addition to bacteria-derived metabolites such as short-chain fatty acids (SCFAs) and indole, which likely induce GLP-1 secretion in colonic L-cells [12,124]. Evidence from in vitro models highlighted that the SCFA propionate stimulates the secretion of PYY and GLP-1 through interactions with free fatty acid receptors 2 and 3 [125]. Human-derived organoid cultures identified that bacteria-derived secondary bile acids such as lithocholic acid stimulate GLP-1 release by activating the G-protein-coupled bile acid receptor 1 [126]. Gut microbiota, such as *Lactobacillus* strains in co-culture with EECs, increase GLP-1 secretion by downregulating the expression of adaptor proteins involved in TLR signaling such as MyD88 and CD14 [127]. This may demonstrate the direct effect of lactic acid bacteria (LAB) on host physiology (Figure 1).

Enterochromaffin cells regulate intestinal peristalsis, enzymatic secretions, inflammatory responses, and are major producers of peripheral serotonin, derived from dietary tryptophan [137,138] (Figure 1). Studies in germ-free mice found that the reduction in serotonin in the blood and colon could be restored through microbial recolonization [139]. For example, several strains of bacteria such as *Escherichia coli*, *Streptococcus thermophilus*, *Lactococcus lactis subsp. cremoris*, *Morganella morganii*, *Lactobacillus plantarum* and Hafnia alvei are reported to produce serotonin in vitro, which contributes to host serotonin levels [12,117,118,119,140]. Additionally, gut microbiota regulate colonic serotonin biosynthesis and increase intestinal motility by releasing enteric neural serotonin through the serotonin receptor 4 (5-HT4, localized in presynaptic enteric neurons) to promote maturation of intestinal neural networks and neurogenesis in the ENS for gut–brain crosstalk [141,142]. *Blatia* spp. appears to modulate host serotonin metabolism by increasing tryptophan hydroxylase 1 expression within the gut. This in turn decreases the relative abundance of microbiota related to intestinal dysmotility, impaired gut barrier function, and psychiatric disorders such as depression or autism [143,144]. The colonization effects derived from *Clostridium ramosum* were attributed to the capability of inducing serotonin production in enterochromaffin cells [145,146,147]. *Clostridium* spp. likely contributes to serotonin secretion by increasing SCFAs and secondary bile acids [139]. *Clostridium* spp. also possess a high 7α-dehydroxylation activity which aids in the production of deoxycholate, thereby promoting serotonin synthesis from enterochromaffin cells [139]. This demonstrates the active role of the gut-associated microbiome in maintaining both epithelial integrity and brain health.

### 2.2. Immunomodulation and Dysregulation

Disorders of the CNS are often accompanied by intestinal inflammation [148,149]. Gut-associated microbiota modulate host immune function and oxidative stress at the local and systemic level [149,150]. The mucosal immune system is composed of the GI tract, lamina propria dendritic cells, gut-associated lymphoid tissue, LP-lymphocytes, and intraepithelial lymphocytes [102]. Hosts tolerate commensal antigens by constantly recognizing and reacting to microbes to ensure the mutualistic nature of the host–microbial relationship [151,152]. Therefore, the immune system is integrated as a bidirectional signaling hub between the CNS and the gut [153]. Microbiota are essential in lymphoid structure development, epithelial, and innate lymphoid cells function, and T-cell subsets at a systemic level. These cells maintain gut mucosa homeostasis by signaling for various proinflammatory (Th1, Th2 and Th17 cells) and anti-inflammatory Foxp3+ Regulatory T-cells (Tregs)) differentiation or response [149,154]. Tregs cells are concentrated in the colon and are influenced by intestinal microbiota [155]. Commensal microbiota likely influence anti-inflammatory Treg cell generation and immune responses through the production of both butyrate and propionate, thereby limiting oxidative-stress-induced inflammation and promoting gut homeostasis [133,149,156] (Figure 1a and Figure 2a). In murine models, the application of the *Bifidobacterium*, *Lactobacillus*, and *Clostridium* genera increase Foxp3 Tregs, which are associated with the protection and attenuation of allergic responses in order to maintain immune homeostasis [157,158,159,160].

PD is associated with intestinal inflammation, in which 80% of patients are observed to have gut microbiota composition and abundance differences compared to non-PD controls [4,169]. For example, an overgrowth of *Enterobacteriaceae*, as well as decreases in *Prevotellaceae* or families associated with SCFAs production, are observed in PD [170,171,172,173,174]. *Enterobacteriaceae* such as *E.coli* induce Th17 inflammatory responses through inflammasome mechanisms, leading to the production of IL-1β. The induction of Th1 and Th17 crosses the BBB, stimulating the microglia and, consequently, promoting brain inflammatory reactions [175,176]. Furthermore, as PD is regularly characterized as a lack of dopamine biosynthesis in various neuronal cells, patients are generally treated withL-DOPA to mitigate the effects of PD. Most recently, berberine supplementation was reported to increase dopamine levels in the brains of mice with PD by modulating the gut microbiota, such as *Enterococcus* spp. [177]. In a study employing metagenomics in PD patients who were not taking L-DOPA, it was found that microbiota composition at the taxonomic level was modified [178]. Specifically, there were predicted functional changes that were associated with intestinal barrier and immune functioning, leading to functional differences in β-glucuronate and tryptophan degradation pathways, which were adverse in those not taking L-DOPA[178].

PD patients display high levels of proinflammatory cytokine expression (TNF-α, IFN-γ, IL-1β and IL-6) and glial activation markers (GFAP and Sox-10) in the colon [179]. TNF-α enters the brain and induces inflammation by activating microglia or astrocytes; oligomeric protofibrils are uptaken and presented to CD4+ by antigen presenting cells [175,180,181] (Figure 2B). Oral administration of *Proteus mirabilis* promotes α-synuclein aggregation in the gut and SN, impairing the colonic barrier, and contributes TNF-α and TLR4-mediated macrophage activation [182]. Thus, the dysbiosis of the gut microbiota induces PD-related pathological changes by increasing gut permeability, promoting TNF-α transit via blood to the SN, as well as triggering α-synuclein aggregation and dopaminergic neuronal damage [175,182,183].

## 3. Heavy Metals and the Gut–Brain Axis

Various nutrients, vitamins, and reactive metals are consumed daily by humans through the ingestion of contaminated foods or water [95,184,185,186,187,188,189]. Simultaneously, the influx of global industrial emissions of heavy metals into environments is linked to environmentally derived brain pathologies; however, preliminary environmental exposures likely interact with the gut-associated microbiome before the brain [190,191] (Figure 3). Once metals enter the GI system, the gut microbiome potentially mediates metal toxicity through biochemical reactions of oxidation or reduction. However, heavy metals promote oxidative stress and disrupt healthy microbiomes in humans, thereby inciting dysbiosis [113,192]. When protective commensal microbial communities enter a state of dysbiosis, there is an increase in the toxic effects of heavy metals [26] and long-term oxidative stress insults, which are in turn associated with neurobehavioral and neurological pathologies [193]. Heavy metal toxicants, which are not modulated by the host-associated microbiome, transit to the brain and can lead to a wide-range of deleterious effects, such as neuroinflammation and neurodegeneration [22].

The clinical and epidemiological characterization of PD suggest that environmental risk factors such as heavy metal toxicity, occupational farming, and pesticide exposure are major factors in PD pathophysiology [194,195]. Heavy metals derived from these environmental exposures lead to the production of reactive radicals such as ROS or NOS that are normally observed in PD progression [38,196] (Figure 3). Specifically, the acute exposure to divalent metal ions incites oxidative stress and oligomerization risk for α-synuclein formation in the brain [197]. Furthermore, Mn, Pb, and Cu ions increase α-synuclein fibrils formation, which produces to Lewy body formation, a major pathophysiological characteristic of PD [54,198,199]. Moreover, the chronic exposure to high levels of metals leads to tissue accumulation over time, especially in the brain [38,200]. Due to the vulnerability of the CNS to metals and the lack of a natural means to detoxify, excrete, or eliminate metals, neuronal energy homeostasis and antioxidant balance are perturbed, which leads to neurodegeneration [63,201,202,203]. This is the case for dopaminergic nerves of the nigrostriatal system, which are readily stressed by heavy metal exposure and may be associated with PD onset [204]. Thus, there is a necessity to further investigate the study of the prodromal phase of PD in relation to metal toxicity pathomechanisms to better understand the underlying disease etiology.

### 3.1. Metal Transport, Movement, and Molecular Mimicry

Environmental exposure to heavy metals alters host metal transporters through molecular mimicry by interacting with nutritional targets, resulting in metal uptake into cells and further accumulation by the host [205]. Uptaken metals incite injury by forming complexes that mimic endogenous molecules such as amino acids or peptides, thereby binding to transporters to reach apical membranes and promote oxidative stress [201,206,207]. For example, within the intestinal lumen, Hg^2+^ forms ligands with S-thiols to stimulate endogenous proteins through molecular mimicry mechanisms. Likewise, CH3Hg-S g mimics GSH, Cys-S-Hg-S-Cys acts as a mimic of cystine; both Cys-S-Cd-S-Cys and G-S-Cd-S g are structurally similar to cysteine and glutathione disulfide [208]. These molecules are then taken up by amino acid transporters, multidrug resistance-associated proteins, or oligopeptides transporters, and then readily trafficked into the brain [54,201,209].

There are a diversity of transporters for the purpose of shuttling endogenic molecules, these are readily employed by heavy metals to exert toxic effects on humans. The best studied is DMT1 which is expressed in the duodenum, erythrocytes, liver, and kidneys; this membrane protein is essential for up-taking divalent metals cations (Pb^2+^, Hg^2+^, Fe^2+^, Mn^2+^) and increasing metal accumulation in tissues [32,206]. For example, human colorectal cells with DMT1 expression knocked-down have a decreased uptake of Fe^2+^, Pb^2+^, and Cd^2+^ [203,206,207,210,211,212,213]. Organic anion transporters that are involved in intestinal uptake of CH3Hg+ through Zinc carriers (ZIP8 and ZIP14) interact with both Hg^2+^ and Mn^2+^. ZIP8 and ZIP14 work in tandem with the zinc-regulated, zinc transporter 1 to transport Cd^2+^, a mimic of Zn^2+^. More so, Cd^2+^ and Mn^2+^ are transported through Ca^2+^ channels and via endocytosis (Figure 4, Table 1) [202,203,207,214,215].

Within intestinal mucosal cells, various metal ions (Zn^2+^, Cd^2+^, Hg^2+^, Pb^2+^, Fe^2+^) cross the apical membrane and readily bind with metallothioneins due to the high content of thiol groups [241,242]. Metallothioneins (MT) have various roles in homeostasis such as the storage, scavenging, transport, and detoxification of free metal ions, thus mitigating cellular damage [242]. For example, MT1 and MT2 are overexpressed in the duodenum when Cd accumulates in intestinal cells [104]. However, Cd promotes the release of Fe from MTs, leading to ROS production and oxidative stress [241,243]. Other examples include the L-system (LAT2), MRPs (MRP1,3,4 and 5), metal transport protein 1 (MTP1), ferroportin, and hephaestin, which participate in the absorption of dietary Fe (Table 1) [201].

Metal ions are able to permeate into the blood through the intestinal barrier by utilizing several transporters, such as transferrin [205,244]. For example, Pb interacts with both hemoglobin and delta-aminolevulinic acid dehydratase (ALAD); however, both Mn and Mg are able to bind to albumin or transferrin and are transported in the blood [245,246,247]. Albumin binds with numerous metals such as Cu, Hg, and Zn [248] (Figure 4). Through these mechanisms, metals are transported to various target organs such as the liver, kidneys, and brain, where they accumulate [184,199,249,250,251]. Metals do not bind to all molecules, rather many mimic molecules, this in turn can lead to an alternative function and oxidative stress. For example, Pb inactivates the heme-associated ALAD enzyme by binding to thiol-groups, leading to ROS production [225]. Cd, Fe, Cu and Mn bind with apolipoprotein A-I, suggesting that exposure to Cd results in oxidative stress by disturbing lipid metabolism [252].

Endothelial cells are a major component of the BBB, in addition to astrocytes, pericytes, basement membranes, and various proteins. Brain endothelial cells enable the process of transportation, such as allowing for the entry of nutrients and other metabolites into the brain [253]. Similar to the intestinal barrier, endothelial cells and neuronal cells within the brain express transporters to acquire biological metals. Specifically, Zn, Fe, and Cu are trafficked by major brain transporters and are readily uptaken across the BBB [207]. Fe acts as a cofactor, leading metals to enter the brain and cross through endothelial cells by transferrin receptors (TfR). The Fe2-Tf complex, which binds to the TfR, is invaginated in order to fuse with endosomes. Eventually Fe is released into the cytoplasm with the support of DMT1. Cytoplasmic Fe is then transferred into the mitochondria or stored in cytosolic ferritin [254,255,256].

Mn binds to Tf and is transported by Fe or Zn transporters into the brain leading to metal accumulation. While the uptake mechanisms are not clear, dietary Fe may influence Mn transport through non-competitive mechanisms [257,258]. Additional carriers found in the BBB, such as ZIP8 and ZIP14, located in the apical and basolateral membranes, both support the bidirectional flux of Mn2+, while ZIP1 and ZIP 2 support the uptake of Pb [207,215]. Additionally, MeHg likely binds to sulfhydryl groups and forms a complex with L-cysteine, which enables transport across the membranes through the L-type amino acid transporter 1 (LAT1). LAT-1 is located in endothelial and pericyte cell membranes and recruits MeHg into the brain; the shuttling of MeHg into the brain further induces the dysfunction of astrocytes and pericytes by increasing BBB permeability [222,259]. Specifically, MeHg exposure inhibits aquaporin AQP4 in astrocytes, leading to alterations in water balance and, consequently, contributing to neurodegeneration [222,260]. Similarly, Pb is taken up into astrocytes through voltage-dependent calcium channels; however, recent studies highlighted the role of Connexin 43 (Cx43) in relation to the uptake of heavy metals such as Pb [261]. Cx43 is a gap junction protein expressed in endothelial cells, astrocytes, activated microglia, and neurons. This gap junction transports metals throughout neuronal cells and is an essential mechanism for disseminating the deleterious effects of metals upon brain cells [261]. Cd uptake is related to Fe uptake mechanisms through DMT-1. Alterations in tight-junction proteins such as ZO-1 translate to uncontrollable Cd uptake in the brain, which in turn induces the impairment of neural tissue functioning [249,262].

### 3.2. Oxidative Stress and Inflammation in the Gut and Brain

ROS (superoxide, hydrogen peroxide, and hydroxyl) produced from metals generate deleterious products which impair key enzymes of the mitochondrial electron transport chain, leading to alterations in protein functions [263,264,265]. For example, the hippocampus, which is associated with memory, is impacted by heavy metals affecting various regions of the brain in PD patients [15]. Metal stress upon the hippocampus is linked to accelerated aging, memory loss, and dementia [266,267]. The hippocampus contains higher than average levels of glutamate and glucocorticoid receptors, which can predispose the hippocampus to metal stress [268].

Heavy metals further induce oxidative stress by generating ROS, which impair tight junctions that regulate barrier and permeability functions of both the GI tract and BBB by modifying phosphorylated junction proteins, thereby inducing chronic inflammation (Table 2) [28,201,269,270,271,272,273]. However, oxidative stress caused by ROS can be counteracted via antioxidants. Across various organisms, glutathione (GSH) mitigates cellular damage by ROS. Specifically, metal ions accelerate hydroxyl radical production through Fenton reactions, leading to GSH reductase inhibition and, eventually, a decrease in GSH concentration [274] (Figure 4 and Figure 5). Specifically, when ROS levels are greater than antioxidant capacity, oxidative stress is generated. Thus, the increase in ROS triggers cell damage through lipid peroxidation, DNA fragmentation, mitochondrial damage, and other cellular alterations that disrupt barrier tight junctions [273].

The human GI tract contains a diversity of metabolites produced by the microbiota. These microbiota metabolites mediate gut–brain communication by transiting to the CNS, thereby modulating oxidative stress and promoting microglia activation [192,295]. Oxidative stress disrupts gut and BBB barriers, leading to α-synuclein misfolding, aggregation, and subsequent neuronal damage in both ENS and CNS [183]. Increased intestinal permeability allows for the translocation of heavy metal ions and bacterial antigens, both are critical in promoting intestinal neuronal oxidative injury. Synthesis of nitric oxide (NO) in the GI tract by inhibitory motor neurons is readily translocated into the brain through the vagus nerve [296]. On the other hand, NO that is produced by the gut microbiota, is scavenged by hemoglobin and is later diffused. This in turn provides free radicals which can interact with α-synuclein, microbes, and immune signaling, eventually leading to a perturbed BBB and neurodegenerative disorders [148].

Microbial dysbiosis of the GI system promotes a pro-inflammatory environment and alters barrier permeability. This factor, in addition to free heavy metal ions which leak from the gut, likely induces downstream oxidative stress in the ENS, leading to α-synuclein misfolding and aggregation [54,297]. Alterations in microbiota stimulate ROS production by activating the cytoplasmic NLRP3-associated inflammasome, regulating the maturation and secretion of pro-inflammatory cytokines, such as IL-1β in epithelial cells, thereby promoting Th17 cell differentiation [156,298]. Gut-mucosa Th17, an inflammatory subset of T-helper cells, are associated with the development of autoimmune disorders, in addition to PD [175,299,300]. Caspase-1-deficient and stressed mice which lack inflammasome activation, elicit a reduction in depressive-like behaviors and have an altered fecal microbiome. The changes in the microbiome are related to beneficial effects, such as rebalancing in gut microbial communities, and the attenuation of inflammation, all highlight that modulation of the gut microbiota via inflammasome signaling likely alters brain functioning [301]. Proinflammatory signaling from LPS and TNF directs nitric oxide synthase (iNOS) upregulation by inducing the oxidation and nitration of the actin cytoskeleton, thereby disrupting the GI mucosal barrier by depleting both occludin and zonula occludens-1 [136,302,303,304]. Due to the fact that the GI submucosal neurons and terminal axons are proximal to the gut lumen, this may lead to the spread of α-synuclein from the ENS to the CNS through the vagus nerve pathway [297]. Furthermore, when dopaminergic nerve cells are exposed to chronic oxidative stress, this insult eventually leads to the characteristic motor symptoms of PD [20,25]. These tandem reactions trigger neuroinflammation and the onset of neurodegeneration [305].

Heavy metal exposure and ROS by-products promote neurotoxic disturbances such as cognitive impairments, learning/memory dysfunctions, movement disorders, loss of language skills, nervousness, and emotional instability [28,204]. Human brains are especially susceptible to fatty acid oxidation due to high rates of oxygen consumption within this organ and ROS stress caused by heavy metals. Specifically, when fatty acids contained in vascular endothelial cells, neurons, and astrocytes are exposed to MeHg, these cells become impaired and cannot uptake antioxidants; this cascade further leads to chronic oxidative stress, inflammation, and endothelial dysfunction [203,306].

Mn exposure has systemic effects on ROS regulation and production, primarily upon enzymes that control both protein folding and transcription [307,308]. A recent RNA-Seq study investigating the acute and chronic effects of Mn on human-derived neuroblastoma cells observed that Mn at any dose incites damages to the nervous system by upregulating oxidative stress, leading to mitochondrial dysfunction and a heightened inflammatory response [68]. Mn prevents neurotoxin clearance by promoting cellular oxidative stress and impairing crosstalk between both neurons and astrocytes by dysregulating calcium signaling, the glutamate–glutamine pathway, and glutamate–GABA cycle [281,295,309,310]. These metabolic disturbances drive oxidative stress, mitochondrial dysfunction by mitigating ATP levels, induce protein misfolding, and eventually, neurotoxicity through neuroinflammation, leading to nigrostriatal cell death [40,281]. Mitochondrial oxidative stress further exacerbates ROS production and drives neuroinflammation in the brain by perturbing muscarinic and dopaminergic receptors [311,312]. Furthermore, the Mn^3+^ oxidation of dopamine increases the relative concentration of localized, oxidized dopamine, resulting in increased oxidative stress [60,313]. While Mn impacts neurotransmitter uptake and trafficking through neuronal oxidative stress causes impairments in cellular metabolism, this pathomechanism is shared between both Mn neurotoxicity and PD onset; however, there is a lack of mechanistic evidence connecting Mn exposure to PD causation [314,315,316,317,318] (Figure 5A).

Inorganic Hg^2+^ that is oxidized by ROS has an increased half-life in the brain and binds to the sulfhydryls of thiols, leading to disruptions in protein conformation or enzyme functions [340,341]. By binding to GSH and the cysteine residues of hormones, essential functions such as GSH upregulation, mitochondrial functions, and the inhibition of the NFkB pathway are blocked, thereby leading to neuronal damage [342,343,344,345]. For example, MeHg causes apoptosis within 18 h of exposure by impairing mitochondrial mRNA expression, inciting the mutation of mtDNA, and leading to the excessive production of ROS [346,347]. When heavy metals damage mitochondria, and GSH concentrations are inhibited, Hg is able to concentrate in the midbrain and promote ROS production prior to PD onset by depleting dopamine production of dopaminergic cells [348,349,350]. This mechanism of neurodegeneration occurs due to the fact that Hg has a high affinity for dopamine receptors as compared to other metals, thus preventing GSH from mitigating ROS and leading to the impairment of numerous neuronal cellular processes and, eventually, hallmark neurological pathologies preceding PD [38,351].

Fe exerts neurotoxic effects through the production of ROS and reacts with hydroxyl radicals; the production of hydrogen peroxide interacts with ferroptosis to release Fe ions that oxidize dopamine and induce dopamine catabolism [352,353,354,355]. The tandem increase in both Fe and ROS in the brain of PD patients is associated with substantia nigra damage [356]. This oxidative stress cascade can result in: neural degeneration, the loss of dopaminergic cells and apoptosis; the overall effect of chronic oxidative stress leads to the exacerbation of neuroinflammation in PD patients [319,320,321]. Chronic inflammation perpetuates Fe ion accumulation in the brain and modifies proteins which metabolize Fe; this further drives neuronal apoptosis by inhibiting mitochondrial complex I [322,357,358,359,360]. Over time, the chronic inflammation by free Fe ions creates ROS, increases the presence of both IL-6 and L-Ferritin in the cerebrospinal fluid, and, finally, directs neuronal apoptosis, which is presented as recurrent tremors [84,361].

BBB dysequilibrium allows for the entry of metals by increasing peripheral inflammatory immune cell infiltration into the CNS, further promoting PD development [362]. Microglia are able to sense diverse stimuli which disrupt CNS homeostasis, such as toxicants, neuronal damage, microbial infection, and α-synuclein [326] (Figure 5c). Initial microglial activation allows for leukocytes to enter the BBB, thereby recruiting Th1 and Th17 cells to produce cytokines that secrete inflammatory molecules such as cytokines, chemokines, reactive free radicals, and others (e.g., iNOS, NO, IL-1beta, IL-6, TNF-α, IFN-γ, IL-8, TGF-b, prostaglandins, cyclooxygenases) [159,160,161]. This in turn may worsen neurodegenerative conditions, inducing neuronal inflammation by stimulating the inflammatory M1 phenotype in the microglia, thereby contributing to neuroinflammation and neurodegeneration [327,328,329,330,331,332,333]. Overall, the chronic activation of M1 microglia is implicated as a key component in the development of neurodegenerative diseases [333].

Astrocyte activation through chronic metal exposure promotes the secretion of pro-inflammatory TNF-α, IL-6, and IL-10, which result in BBB tight junction impairment [363,364,365] (Figure 5e). The protective role of astrocytes, which effectively clears α-synuclein and neurotrophic factors, is impaired following metal alterations of glycogen consumption by inducing ROS production, further promoting astrogliosis and dopaminergic neuron degeneration [366,367]. This mechanism is essential in PD onset due to the fact that if α-synuclein is overproduced and taken up by astrocytes, this likely leads to astrogliosis and neurodegeneration [339]. Thus, the overactivation, abnormal influx, and loss of regulatory activities within astrocytes is a significant contributing factor in the progression of immune-mediated PD [338,339].

### 3.3. Microbes and Heavy Metals in Gut–Brain Bioremediation

Minerals and trace elements (Ca, P, Mg, Fe, Cu and Zn) are essential for intestinal homeostasis and absorption [368]. However, diets contaminated with heavy metals can result in adverse effects through a variety of mechanisms [369]. The transport of nutrients occurs in the small intestines; the contact or accumulation of metal ions within the GI epithelium induces oxidative stress, cellular injury, dysbiosis, and increases the abundance of facultative anaerobes [207,370]. As a consequence, the available epithelial oxygen increases, leading to the depletion of anaerobic SCFA-producing microbiota, thus reducing the production of anti-inflammatory or antioxidant metabolites. In a healthy environment, oxidative stress is mitigated through chelation or the microbial-mediated reduction, uptake, and eventual clearing of heavy metals (Table 3) [99,371].

Chronic exposure to toxic metals, such as Cu, Pb, Zn, and Cd, alter the gut microbiome of individuals who reside in metal-polluted environments by shifting gut-associated microbiota towards a relatively high abundance of *Lachnospiraceae* spp., *Eubacterium eligens*, *Ruminococcaceae UGG-014*, *Erysipelotrichaceae UCG-003*, and *Tyzzerella* spp., as well as significantly decreasing *Prevotellaceae* [399]. Overall, this change in microbial composition by metal exposure incites metabolic changes in gut microbiota, affecting host metabolism. A stable and functional gut-associated microbial community is necessary to remediate xenobiotic metals. Gut microbiome commensals induce numerous factors which protect hosts from metal stress through the expression of endogenous metallothioneins [400], the upregulation of glutathione (GSH) for anti-oxidative activities [401], and other relevant processes [105]. These factors act in orchestration to limit heavy metal uptake to modulate metal-associated pathogenesis. For example, gnotobiotic mice exposed to Cd and Pb via oral ingestion, have a significant 5–30 and 7–40-fold increase in metal accumulation compared to controls in fecal samples, respectively [27]. These results suggest that host-associated microbiota are essential barriers to mitigate and bioremediate heavy metals in the gut and brain.

A broad group of anaerobic digesting microorganisms described as LAB are key colonizers of the human GI tract. LAB generally act as probiotics and human symbionts, which lack deleterious effects on human health. LAB tolerates acids, degrades different carbohydrates, binds metals to decrease intestinal metal absorption, and reduces tissue metal accumulation, thereby mitigating oxidative stress [402,403]. LAB metal bioremediation properties also create a strong competition in human-associated niches as probiotics [106,392,404]. Human gut LAB likely prevents and bioremediate metal toxicity associated with neurodegenerative disease in at-risk populations [405,406]. LAB responds to various dietary heavy metals, and this is indicative of a potential intervention strategy as a neurological probiotic for the clearance of metals (Table 4).

Probiotic *Bifidobacterium* and *Lactobacillus* are found in human infants and increase both gut microbiota diversity and healthy immune functioning [415,416]. *L. rhamnosus* GR-1 mitigates metal translocation by employing mechanisms of heavy metal tolerance by binding, sequestering, and clearing excess metals for the host [417]. Organic acids produced by LAB such as *Lactobacillus*, chelate toxic metals by decreasing pH and increasing metal solubility, resulting in the formation of metallo-organic molecules [418,419,420]. This occurs through three mechanisms: ion exchange, precipitation through nucleation reactions, and by creating metal nanoparticles in bacteria cell walls [24,421,422,423].

Probiotic LAB uniquely ferments complex fibers to produce SCFAs (acetate, butyrate, and propionate), thereby providing nutritional supplements for colonocytes to support gut barrier maintenance and prevent invasive pathogens [125,424]. Cd-exposed mice are observed to have a decreased abundance and growth of the genera *Lactobacillus* and *Bifidobacterium*, resulting in an impaired GI barrier and Cd accumulation [425]. Low butyrate production increases pH in the gut, creating conditions favoring the overgrowth of invasive pathogens [91,107,164]. Thus, decreased SCFA production promotes increased gut permeability, the growth of opportunistic bacteria, increased uptake of metals, and triggers chronic inflammation [425,426]. When SCFAs are limited, both intestinal dysbiosis and chronic inflammation are heightened [104]. The oral administration of *L. plantarum* is beneficial in the remediation of Cd, Pb and Cu toxicity; this mechanism inhibits metal uptake by intestinal cells and allows for metal excretion in the feces [389,427,428]. *L. rhamnosus* GR-1 (LGR-1), which is supplemented in yogurt, demonstrated a decrease in the levels of both As and Hg in the blood of pregnant women and children. These findings suggest that various *Lactobacillus* strains are potential therapeutic candidates in the remediation of heavy metals to prevent diverse diseases, such as neurodegenerative pathologies [429].

LAB strains also synthesize branched-chain amino acids (BCAAs). These amino acids modulate host physiology by increasing mitochondrial biogenesis, leading to increased antioxidant effects against ROS, and are able to penetrate the BBB [430,431,432,433]. Low plasma BCAAs were associated with autism; however, dietary supplementation with BCAAs reverses some neurological phenotypes [434]. With regard to PD, recent studies identified that PD patients generally have reduced concentrations of both BCAAs and aromatic amino acids compared with controls [435,436], while other studies observed increased BCAA concentrations in PD patients as compared to controls [437,438]. These conflicting observations on the effects of BCAAs, which are derived from LAB, elicit the need to further investigate the mechanistic effect of BCAA upon neurodegeneration and PD onset.

LAB supports the production of various gut-associated secondary metabolites, which can potentially influence neuronal health. For example, *L. rhamnosus* JB-1 was successfully employed as a therapeutic probiotic to reduce stress, anxiety, and depressive behaviors in mice [439]. Furthermore, decreases in microbiome-derived acetate in mice treated with the antibiotic vancomycin demonstrates a reduction in synaptophysin in the hippocampus, leading to impairments in learning and memory [424,440]. Propionate from LAB interacts with FFAR3 in the ANS, PNS, and CNS to induce intestinal gluconeogenesis [424]. Probiotic strains such as *Lactobacillus*, *Bifidobacterium*, and *Bacteroidetes* are also a significant source of bile salt hydrolase; this allows for the deconjugation of bile acids from taurine and glycine [12,441]. Recently, it was hypothesized that bile acids modulate neuronal function through the stimulation of Fibroblast Growth Factor Receptor (FGF)15 secretion by Farnesoid X receptor (FXR) in the GI, silencing hypothalamic Agouti-related protein/neuropeptide Y neurons through FGF receptors [442].

A primary inhibitory neurotransmitter in the brain, GABA is produced by both the host and GI-associated microbiota [443]. Alterations in GABA and glutamate metabolism were identified during Mn accumulation [444]. The neurotoxic effects include effects in the dopaminergic function in the striatal zone on the brain, where high rates of both oxygen consumption and metabolic rates are crucial factors for increasing brain sensitivity to oxidative damage. [444,445]. Additionally, alterations in GABA receptor expression are related to anxiety and depression symptoms, which occur in tandem with IBD [147,446]. Mice supplemented with probiotic *L. rhamnosus* were found to have decreased anxiety levels in addition to increased gene expression of both GABAAα1 and GABAAα2 in the hippocampus[432]. Histamine, synthesized by both microbiota and epithelial gut cells of the host have essential roles for mediating gut microbiome and host neurocommunication [447,448]. Application of the probiotic *L.reuteri* activates histamine receptor 2, thereby modulating host mucosal responses to inflammation and highlighting the role of probiotic microbiota in gut-brain communication [449].

## 4. Systems Toxicology

### 4.1. Towards Systems Toxicology

In our day-to-day life, humans are exposed to an environmental cocktail of heavy metals through their diet. Conversely, classical toxicity testing and the risk assessment of metals typically investigates a singular exposure. Therefore, the consideration of metal mixtures upon multiple target organ systems is lacking in detail. Specifically, the application of “Omics” methods can be utilized to increase the mechanistic understanding of metal mixtures toxicity upon biological pathways such as oxidative stress, mRNA splicing, and ETC dysfunction in relation to neurodegeneration and PD [259,450]. As extensively discussed in previous sections of this review, heavy metals toxicity influences neurodegeneration via the microbiota through a wide variety of pathomechanisms. The utilization of different “omics” technologies such as transcriptomics and metagenomics, needed to generate large “molecular snapshots” of various conditions underlying neurodegeneration in the gut–brain axis, are essential [451,452]. While these approaches provide the means to obtain sizable quantitative datasets describing a variety of molecular functions that underlie neurodegeneration, a major hurdle still exists: the data-driven integration of multi-omics and microbiome data into a large-scale system toxicology analysis through modeling approaches [453].

Systems biological knowledge bases and models can be readily employed to integrate and interpret these data at a systems level, providing mechanistic hypotheses about the interactions between the large variety of biological components involved. For example, the alterations that metal toxicants induce between different biological processes, such as oxidative stress responses, mRNA splicing, and energy metabolism and their relation to neurodegeneration [450,454] can be characterized at the levels of genes, proteins, and metabolites. Systems biology models can be employed in this case to infer systems-level activity changes to characterize the interplay between metal toxicants, gut bacteria, and human tissues. In the next section, we introduce a metabolic modeling approach which is specifically designed to simulate the metabolic interactions occurring between multiple human organs, including the gut and its bacterial community. We will speculate about how this approach can be employed to study the role of the microbiota in modulating heavy metal uptake and toxicity effects.

### 4.2. Whole-Body Models Integration with Heavy Metals Bioremediation

Various systems biology tools such as Constraint Based Reconstruction and Analysis (COBRA) [455,456] have proved their worth by deciphering the metabolic activity of single organisms, as well as multi-species metabolic cross-talk. One of the most frequently employed COBRA methods is flux-balance analysis (FBA) [457], which is used to computationally estimate and characterize biochemical reactions activity in reconstructed metabolic networks of the studied organisms under metabolic steady-states. This framework requires the specific definition of a growth environment (e.g., which metabolites are taken up by the modeled organism), which is dependent upon molecules that are known to be available within the environment under investigation.

Over time, these applications have led to the development of whole-body metabolic models (WBM models) [458]. These models represent multiple individual tissues of an organism by their respective metabolic network along with their unique metabolic connections (e.g., via the bloodstream or the lymphatic system). Additionally, previously published WBM models also contain a specific colon luminal compartment with metabolic networks for the gut microbiota [458]. Moreover, depending on the specific questions to be answered, physiological constraints can be integrated into these models to include information such as blood flow and kidney filtration [458]. These applications led to the recapitulation of interorgan interactions such as the lactic acid and alanine/glucose-alanine cycles through simulations by using the WBM approach [458]. Overall, WBM models highlight a promising approach to predict the interactions between multiple organ systems and microbes within specific contexts, which can include toxic exposure as an important factor for the emergence of PD and related diseases.

Additionally, heavy metals have a detrimental effect on the bacterial community [459], which is important for maintaining gut–brain axis homeostasis [12]; microbial communities are, therefore, strong mediators in the pathophysiological balance between health and disease statuses. While the influence of heavy metals on gut-associated microbiota in directing changes in microbial community composition were studied experimentally [107,460], computational studies which propose metabolic interventions to revert those changes are still required. For example, a list of potentially beneficial metabolites could be generated by simulating the effects of several metabolic interventions on the structure of microbiomes altered by heavy metals. This would allow for the screening of specific metabolites that are able to revert a dysbiotic, host-associated microbial composition back towards a healthy state.

In the context of heavy metal toxicity that is associated with neurodegeneration and PD, WBM models could be employed to investigate the extent to which microbial communities could either mitigate or exacerbate brain toxicity effects. It is already known that some specific bacteria or a combination of them, as shown in Table 3, are able to reduce the toxicity of several contaminants; *Streptomyces werraensis* for example is known to absorb hexavalent chromium [461]. The combined effect of multiple bacterial species and their communication within host tissues could be studied through the WBM models approach. Additionally, as WBM models allow to account for sex-specific differences, it will be possible to investigate whether sex-specific trajectories of heavy metal toxicity on the host microbiome [370,462] are related to differences in host metabolism. Other potential applications of metabolic modeling approaches include the assessment of the mechanistic bases which underlie complex biological relationships, such as the effect of gut microbiome taxonomic changes and the influence of IBD treatments upon PD incidence, such as the case for anti-TNF therapy [22,178]. A list of metabolic-dependent heavy metal uptake and transformations pathways that can be employed by the bacteria to alter heavy metals availability in the surrounding environment is extensively explained in Chandran et al. 2020 [406]. An important example of these transformations is the methylation of heavy metals, which increases their solubility, and thus the potential transport to the brain. Specific models that include heavy metal metabolism need to be established, and such models’ creation and validation is of paramount importance for systems toxicology study and will require a large amount of work. The use of models specifically curated to include this information will allow for the estimation of microbiome-specific metal transformation capabilities. Those values could be compared to investigate whether the activity of heavy metals transformation mechanisms correlate with PD severity. Overall, the integration of a gut-associated microbial compartment can allow for the further investigation of host–gut microbial co-metabolism within a systems toxicology context. To provide some examples of similar concepts on a topic that differ from heavy metals and PD, it could be important to take inspiration from systems biology. For example, results from integrated WBM models illustrated that the presence of specific host microbiota does not only bolster the production of essential neurotransmitters, but also recapitulates gut–liver alcohol exchange and colonocytes uptake of bacterial butyrate [458], the latter being essential in maintaining the homeostasis of colon epithelial cells. Moreover, this framework allowed for the modeling of the competition between bacterial metabolites and drug detoxification in the liver [458]. Overall, these results suggest that important metabolic exchanges between microbiome, gut, and brain can be modeled with this approach and could be specifically tailored to include information about heavy metal trafficking throughout the gut–brain axis.

## 5. Conclusions: Fermented Functional Probiotics and Oxidative Stress

Industrialization and urbanization contribute to the release of heavy metals in food and water. Contamination with heavy metals is a serious ecological problem for humans and animals. Although metals are biologically important, they are usually required in trace amounts, excessive metal accumulation in various organs induces various detrimental intracellular events (oxidative stress, mitochondrial dysfunction, DNA fragmentation, protein misfolding, endoplasmic reticulum (ER) stress, autophagy dysregulation, and the activation of apoptosis). Metal accumulation in tissues is known to disrupt the homeostatic functioning of biological systems [405,419]. Thus, harmful compounds can cross organismic barriers leading to alterations within microbiome–gut–brain axis communication; chronic metal toxicity promotes oxidative stress of the gut and brain, eventually leading to PD-associated pathologies [27,38].

The gut barrier, as the first line of defense, largely depends on microbiome–host interactions to control the entry of ingested toxicants. Recent studies illustrated the gut microbiome’s role in modifying the bioavailability and toxicity of heavy metals. Specifically, microbial communities were shown to decrease metal accumulation in both the blood and organs [105]. In this regard, the integration of both in vitro or in vivo studies of toxic exposures with computational modeling allows for the study of community composition, microbiome-specific metals transformation capabilities, gut–brain communication and neurotoxicity from a systems toxicology point of view. This is important due to applications of WBM models that include the prediction of interactions between multiple organ systems and microbes. Tailoring WBM models to account for chronic toxic exposures within specific contexts can facilitate the identification of the mechanistic foundation which underlies these interactions. Moreover, the utilization of novel modeling approaches such as WBM models can be employed to screen for metabolites that appear to be important in the gut-associated microbiome structure and improve the bioremediation potential of microbes, allowing for new strategies to decrease metal uptake and accumulation to improve human health. Thus, potential bioprocessing capabilities of LAB can be estimated as probable interventions to alleviate oxidative stress and metal toxicity, which are significant causative agents of neurodegeneration and the onset of PD.

## Figures and Tables

**Figure 1 antioxidants-11-00071-f001:**
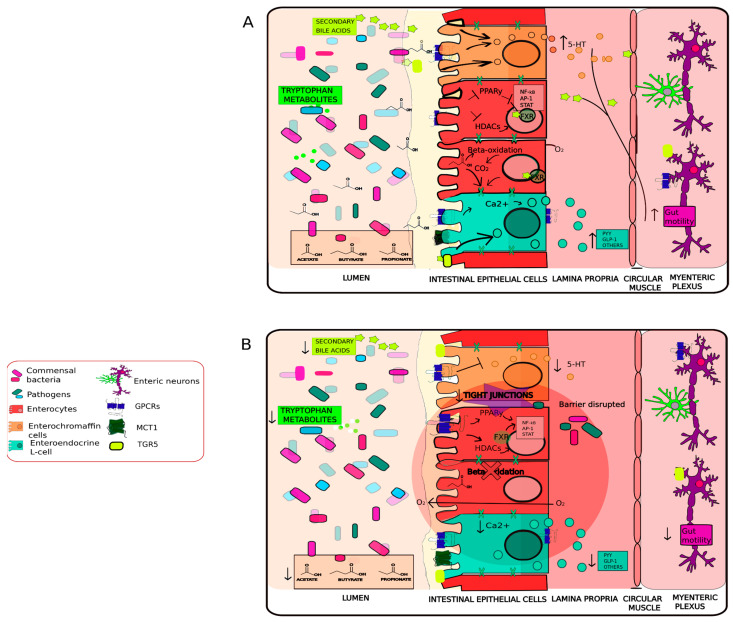
Bacterial metabolites signaling in the intestine. (**A**) Secondary bile acids, SCFAs and tryptophan metabolites are microbially derived metabolites. Secondary bile acids activate FXR and TGR5, which stimulate different types of cells. Enterochromaffin cells are stimulated promoting 5-HT release and motility in the colon. Enteroendocrine L-cells promote glucose tolerance through release of the incretin GLP-1. Enteric neurons with TGR5 stimulate or inhibit motility [128,129,130]. Bacterial fermentation of dietary fibers leads to the reduction in luminal pH and production of SCFAs [131]. The bacterial-produced SCFAs activate G-protein-coupled receptors (e.g., GPR41, GPR43 and GPR109) on enteroendocrine cells, enterochromaffin cells, and enteric neurons, leading to the increased production of GLP-1 and 5-HT, thus directing changes in gut motility [132]. With regards to colonocytes, anionic-SCFAs enter the cell through a carrier-mediated transport (similar to MCT1), promoting of mitochondrial fatty acid beta-oxidation and reduction in luminal availability of oxygen through PPARγ activation [131,132]. Finally, the reduction in oxygen availability upregulates the expression of tight junction proteins which are important for the maintenance of the gut barrier [132,133]. (**B**) Under dysbiosis, there is a limited availability of SCFAs, leading to a decreased amount of available substrate for the colonocytes. This leads to a decreased activation of PPARγ and less oxygen uptake in colonocytes causing dysregulation of cellular growth and differentiation, which can be described as a metabolic switch towards anaerobic glycolysis and lactate production. When the concentration of oxygen increases, decreases in the expression of HIF1 lowers tight junction expression, thus further degenerating the gut barrier [131,132,133,134,135,136]. Furthermore, pathogenic interaction with the impaired cells increases their translocation to the lamina propria, thus promoting oxidative stress events and chronic inflammation [133,134,135,136]. Abbreviations: 5-HT, 5-hydroxytryptamine; FXR, farnesoid X receptor; GI, gastrointestinal; GLP-1, glucagon-like peptide-1; HIF1, hypoxia-inducible factor 1; PPARγ, proliferator-activated receptor gamma; TGR5, Takeda G protein-coupled receptor 5.

**Figure 2 antioxidants-11-00071-f002:**
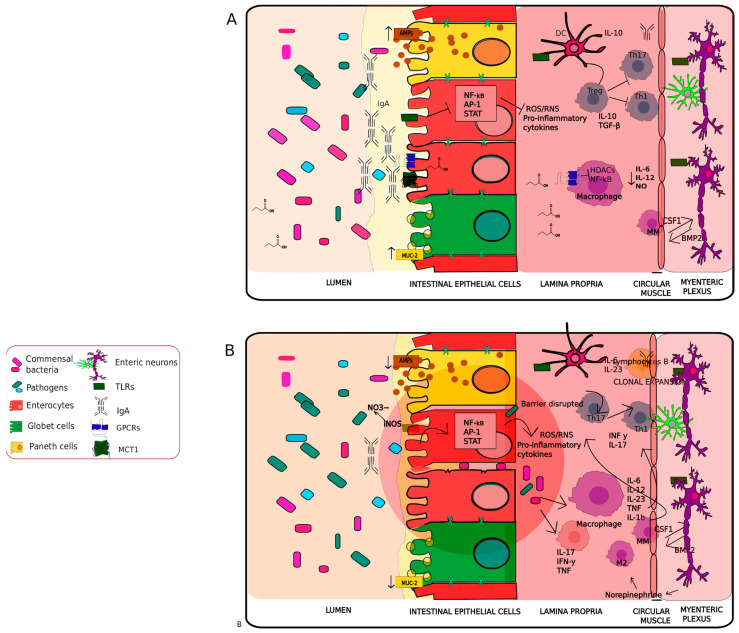
Neuronal and immunity crosstalk. (**A**) Under homeostasis, macrophages mature within the intestinal mucosa, where they are utilized to capture invading pathogens, recognize antigens, and clear old apoptotic cells within the area. Macrophages then transfer the captured antigens to the dendritic cells, which are responsible for entering the mesenteric lymph nodes to induce the differentiation of peripheral T regulatory (Treg) cells from naïve T cells[161]. These cells constitutively produce IL-10, which promotes the secondary expansion of regulatory T cells in the mucosa and maintains the homeostasis [161,162]. Butyrate is important in the modulation of intestinal macrophages via the inhibition of histone deacetylases and NF-kB, and thus downregulates proinflammatory responses such as IL-6, IL-12 and NO [163,164]. On the other hand, neuron enteric cells are essential for gastrointestinal motility. The Toll-like receptors (TLRs) expressed by enteric neurons can recognize gut microbe-derived signals and influence the gut motility[165]. These cells promote crosstalk with the Muscularis Macrophages (MMs) in longitudinal and circular smooth muscles and incite the production of the growth factor bone morphogenetic protein 2 (BMP2) to stimulate motility that is directed by the neurons. In turn, the enteric neurons through the production of macrophage survival factor CSF1 promote the maintenance of the MM [152,165,166]. (**B**) Under dysbiotic conditions, the epithelial barrier is impaired; this in turn leads to the translocation of microbiota, toxins, and deleterious exposures, as suggested by Braak’s Hypothesis, thereby leading to the activation of the host immune response. Macrophages then produce proinflammatory cytokines and present antigens to dendritic cells; together, both cells induce the expansion of TH1 and TH17, thereby recruiting other innate effector cells such as neutrophils and eosinophils [162,163,164]. Additionally, excess TLR activation by the microbiota or toxins, activates cytokine release, proinflammatory responses, and apoptosis in enteric neurons. Thus, MMs decrease the expression of BMP2, and enteric neurons decrease the expression of CSF1 and MM motility[152,166]. Enteric bacteria can further influence the development and differentiation of CD4+and CD8+ T cells, as well as B cell activity and IgA production [167,168]. Abbreviations: BMP2, bone morphogenetic protein 2; CSF1, colony stimulating factor 1; Interferon γ (IFN-γ); TLRs, Toll-like receptors; TNF-α, tumor necrosis factor-alpha.

**Figure 3 antioxidants-11-00071-f003:**
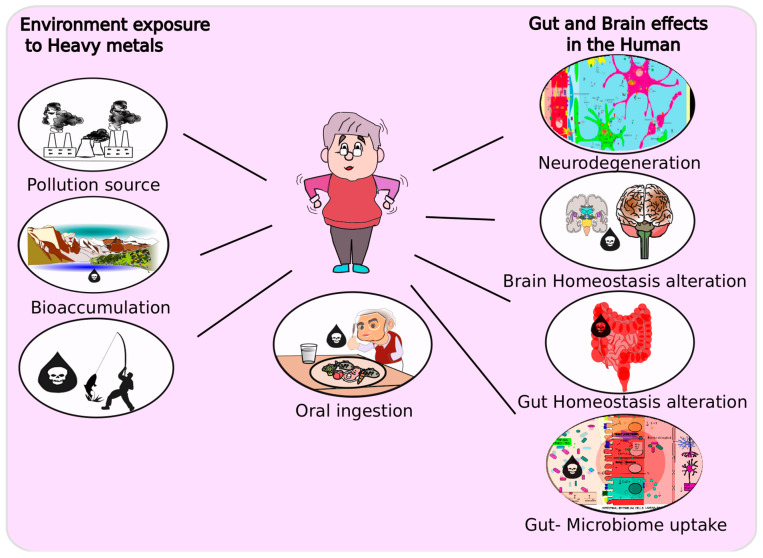
Human exposure to heavy metals in the diet underlies neurodegenerative diseases. Multiple levels of environmental exposure to heavy metals, which are derived from pollution, lead to adverse effects on the gut and brain in humans.

**Figure 4 antioxidants-11-00071-f004:**
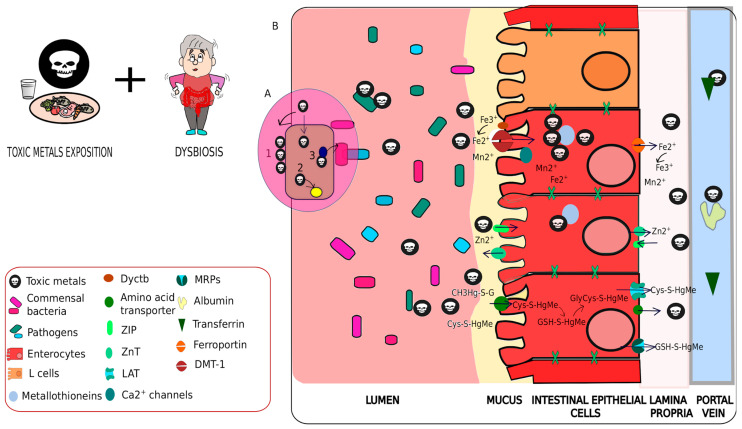
Gut–microbiome metal uptake. (**A**) Mechanisms of microbial heavy metal bioremediation in the gut: 1. Biosorption 2. Bioaccumulation, 3. Biotransformation [216]. (**B**) Schematic of intestinal metal transport: The primary mechanisms by which iron and other divalent metals (Pb^2+^, Hg^2+^, Fe^2+^, Mn^2+^) are taken up by the enterocyte is through DMT-1 from the luminal membrane[217]. In the case of Fe^3+^, this metal is reduced to Fe^2+^ by *DYCTB* [217]. Other receptors for Zn^2+^ (ZIP, ZnT) [201,218,219,220,221,222], Ca^2+^ and amino acids transporters (b0,+, PepT1) [201,223,224] are utilized to uptake MeHg or other metal ions. Iron was also suggested to be transported with heme with HCP1 and as ferritin [225]. Once entering the intracellular space, metals are readily stored as ferritin (Fe^2+^) with metallothioneins and exported into the body circulation through ferroportin or other transporters (LAT, ZIP, ZnT) [201,218,219,220,221,222,226,227,228]. Ferritin is converted to Fe^3+^ by ferroxidase, which is then released for use by transferrin, other metals are transported by albumin or bound to other cell-derived proteins [201,229]. Abbreviations: HCP-1, Heme carrier protein; Znt, Zinc transporter; ZIP, RT, IRT-like protein; LAT, L-type/large neutral amino acid transporter; MRPs, Multidrug Resistance-Related Protein; DMT-1, Divalent metal transporter 1; DYCTB, Duodenal cytochrome B.

**Figure 5 antioxidants-11-00071-f005:**
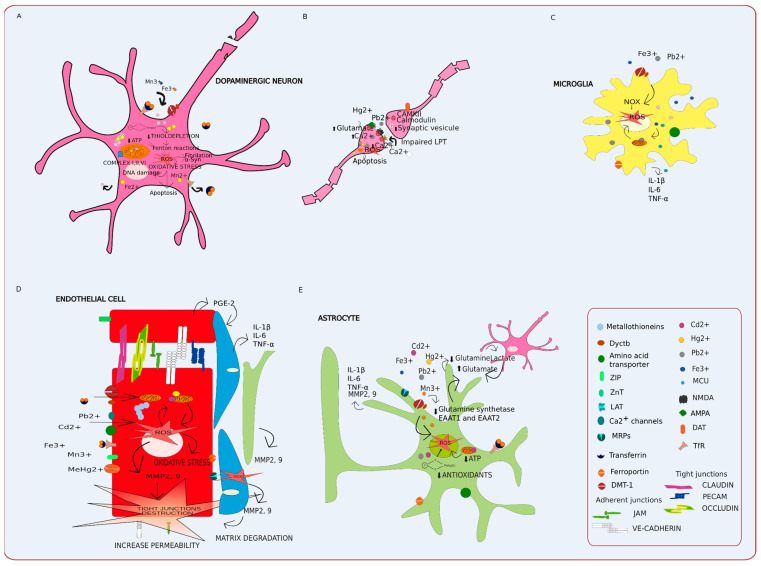
Heavy metals and oxidative stress in neuronal cells: the disruption of BBB through metal neurotoxicity and molecular mimicry. (**A**) Mn, Fe and other toxic metals accumulate within the mitochondria by utilizing the MCU. An increase in oxidative stress insults leads to a dysfunction of the mitochondrial electron transport chain. Metals that are able to bind to ADP via ATP-synthase uniquely uncouples oxidative phosphorylation and ATP formation within the mitochondria. Pb2 has an inhibitory effect on the activity of calmodulin and, consequently, leads to the avoidance of synaptic vesicles release [311,312,313,314,319,320,321,322]. (**B**) In the neuronal synapses, PB, Cd, and Mn impair Ca2+ channels (voltage-gated Ca2+ and Ca2+ ATPase) leading to Ca2+ ion avoidance and alterations in ion signaling. Pb2 has an inhibitory effect on the activity of calmodulin and consequently leads to the avoidance of synaptic vesicle release [323,324]. In the binding sites of postsynaptic cells at the NMDA-receptor/channel, and AMPA disturbances of the postsynaptic potential, leads to the synaptic plasticity and induction of LTP[323,324,325]. (**C**) Glial cells further increase the production of ROS by mitochondrial damage. Specifically, alterations in antioxidant levels and metal reactions further promote the activation of the NF-kB pathway by the release of proinflammatory cytokines (TNF-α, IL-1, IL-6). Increased cellular damage associated with the accumulation of metals promotes α -synuclein formation in astrocytes leads to the suppression of protective functions and a decrease in glutamate uptake, leading to excitotoxicity in neurons [326,327,328,329,330,331,332,333]. (**D**) In endothelial cells, the increase in ROS production promotes the release of metalloproteinase-9 and PGE-2 by endothelial cells. This factor then stimulates both pericytes and glial cells to further promote pro-inflammatory signals and, as a consequence, degrades the extracellular matrix of both tight junctions and BBB permeability [41,273,283]. (**E**) The interaction of glial cells with ROS leads to mitochondrial damage and the release of proinflammatory cytokines, which are further bolstered by the additional burden of heavy metals. This impairs the essential roles of neuron maintenance by disrupting glutamate/GABA-glutamine shuttling. The expression of EAAT1 and EAAT2 and the activity of glutamine synthetase can be downregulated by the excess of intra-astroglial heavy metals [334,335,336,337]. Thus, glutamine catabolism and elevated extracellular glutamate further induces excitotoxicity and neuronal damage, finally leading to neurodegeneration[334,338,339]. Abbreviations: AMPA: α-amino-3-hydroxy-5-methyl-4-isoxazolepropionic acid, DMT1: Divalent metal transporter 1, DAT: dopamine active transporter, EAAT: excitatory amino acid transporter, IL-1: Interleukin-1, IL-6: In-terleukin-6, IL-12: Interleukin-12, JAM: junctional adhesion molecule, LTP: Long term potentiation, MCU: mitochondrial calcium uniporter, MMP-9/-3: metalloproteinase-9/-3, MRP: Multidrug resistance-associated proteins, MTs: metallothioneins, NMDA: N-Methyl-D-Aspartic acid, NADPH oxidase: nicotinamide adenine dinucleotide phosphate oxidase, NF-κB: nuclear factor kappa, PGE-2: Prostaglandin E2, PECAM: Platelet/endothelial cell adhesion mole-cule-1, ROS: Reactive oxygen species, TfR: Transferrin receptor, TNF: Tumor Necrosis Factor, VE-CADHERINE: vascular endothelial cadherin, ZIP: Zinc-imidazolate polymers, ZnT1: zinc transporter protein-1.

**Table 1 antioxidants-11-00071-t001:** Molecular mimicry of metals throughout the gut–brain axis.

Transporters	Molecule/Ion Metal Being Mimicked	Metal Replacement	Cells Containing the Transporters of the Transporter	Citation
Organic anion transporters:		CH3Hg^+^	Endothelial cells/Glial cells/Enterocytes	[201,230,231]
OAT1	GSH			
	Cysteine	GS-Cd-S-G		
OAT3				
		CH3Hg-S-Cys		
Zinc-regulated zinc transporter 1 (hZTL1)	Zn ^2+^	Cd^2+^	Enterocytes/Neurons	[218,219,220]
Ca^2+^ channels	Ca^2+^	Cd^2+^	Neurons/Endothelial cells/glial cells	[201,223,224]
		Pb^2+^		
		Mn^2+^		
Divalent metal transporter 1	Fe^2+^	Pb^2+^	Enterocytes/Endothelial cells/Neurons/Glial cells	[217]
DMT-1		Mn^2+^		
		Cd^2+^		
Zinc-imidazolate polymers (ZIPs)	Zn ^2+^	Pb^2+^	Enterocytes/Endothelial/Astrocytes	[201,221,222]
1.2		Mn^2+^		
8, 14		Cd^2+^		
		Hg^2+^		
Transferrin receptor	Fe^2+^	Mn^2+^	Neurons/Endothelial cells/Glial cells	[229]
TfR				
Amino acid transporters (system b0,+, system L)	Cysteine	CH3Hg-S-Cys	Enterocytes/Endothelial cells/Glial cells	[201,226,227,228]
	Methionine	Cys-S-Hg-S-Cys		
		Cys-S-Cd-S-Cys		
		CH3Hg-S-CysGly		
Multidrug resistance-associated proteins		CH3Hg^+^	Enterocytes/Endothelial cells/glial cells	[201,232]
MRP 1, 2, 3,4	GSSG	G-S-Cd-S-G		
	GSH	CH3Hg-S-G		
		As (III)		
		As–GSH		
Ferroportin	Fe^2+^	Cd^2+^	Enterocytes/Endothelial cells/Neurons/Oligodendrocytes/Astrocytes	[201,233,234,235]
		Mn^2+^		
Glucose permeases	Glucose	As (III)	Enterocytes/Endothelial cells/Astrocytes	[210]
GLUT 1, 2, 5				
Sodium-dependent phosphate transporters	Phosphate	As(V)	Enterocytes	[218,219,220,236,237]
NaPiIIb				
Aquaporins	Glycerol	As(III)		[218,219,220,238,239,240]
AQP 3, 10	Water	Hg^2+^	Enterocytes/Enteric neurons/Endothelial cells/Astrocytes	
AQP 4		Pb^2+^		
		Fe^2+^		
		Mn^2+^		
Organic anion transporting polypeptides	Amphipathic organic compounds	As(III)	Enterocytes/Astrocytes/Endothelial	[218,219,220]
OATPB				

**Table 2 antioxidants-11-00071-t002:** Metal ions and ROS/NOS adversely oxidize neuroproteins and lipid metabolism.

Metal Ion	Reactive Species	Oxidized Molecules	Adverse Outcomes	Cite
Lead	Pb^2+^	δ-aminolevulinic acid dehydratase (ALAD)	Reduces the antioxidant (glutathione) levels	[275,276]
	Hydrogen peroxide Superoxide radical	Superoxide dismutase (SOD)	Oxidative stress	
	Hydroxyl radicals	Catalase	Alteration in Ca^2+^ influx	
		Glucose-6-phosphate dehydrogenase (G6PD)	Apoptosis	
		GSH		
		Glutathione reductase		
		Glutathione peroxidase		
		Glutathione s-transferase		
		Voltage-gated calcium (Ca^2+^) channels		
		N-methyl-d-aspartate (NMDA)		
Cadmium	Cd^2+^	Thioredoxin	Depresses antioxidants (glutathione) levels	[258,274,277,278,279]
	Superoxide anion	Cysteine	Oxidative stress	
	Hydrogen peroxide	Ubiquitin enzymes	Damage in the electron transport chain	
	Hydroxyl radicals	Mitochondrial Complex II III	Apoptosis	
		Topoisomerase II	Lipid peroxidation	
		DNA methyltransferases	Alteration in maintaining genomic integrity	
		GSH		
		Glutathione reductase		
		Glutathione peroxidase		
		Glutathione s-transferase		
Mercury	MeHg	DNA	Depresses antioxidants (glutathione) levels	[280,281,282,283,284,285]
	Hg^2+^	Thioredoxin reductase	Oxidative stress	
	Hydrogen peroxide	Nitric oxide synthase	Lipid peroxidation	
	Nitric oxide	Monoamine oxidase	Mitochondrial function	
		Glutathione reductase	Decreases GABA signaling	
		Glutathione peroxidase	Neurotransmitter metabolism	
		Astrocytic glutamine transporter	Glutamine uptake	
		Choline acetyltransferase	Acetylcholine synthesis	
		Creatine kinase	Decrease ATP production	
		Cytosolic phospholipase A2	Membrane damage in Endothelial cells	
		Enolase		
		Glutamate transporters		
		Ca^2+^ ATP		
Mn	Mn^2+^	Dopamine	Impairment of oxidative phosphorylation	[276,281,286,287,288,289]
	Mn^3+^	SOD	Decrease ATP synthesis	
	DA-o-quinone	Complex I, II	Disruption of mitochondrial energy production	
	Aminochrome	Aconitase	Alteration in Ca^2+^ influx	
		Adenylate cyclase		
		Succinate		
		Malate		
		Glutamate		
		N-methyl-d-aspartate (NMDA)		
		ATP synthase		
Iron	Fe^2+^	Complex I III	Lipid peroxidation	[281,290,291,292]
	Fe^3+^	SOD	Dopamine metabolism	
	Hydrogen peroxide Superoxide radical	α-synuclein	Mitochondrial functions disruption in ATP synthesis	
	Hydroxyl radicals	Tyrosine hydroxylase	Apoptosis	
	3,4- dihydroxyphenylacetaldehyde DA-o-quinone	Hydrogen peroxide	DNA/protein degradation	
	6-hydroxydopamine	Lipid peroxide		
	Aminochrome	DNA/RNA		
		Creatine kinase BB		
		Cytochrome c oxidase		
		Ketoglutarate dehydrogenase		
Arsenic	iAsV	ATP synthase	Uncouples oxidative phosphorylation	[250,293,294]
	iAsIII	β-tubulin	Decrease ATP formation in the mitochondria	
	Monomethyl arsonous acid (MMAIII),	Glucose 6-arsenate	Inhibition of the hexokinase	
	Monomethylarsonic acid (MMAV)	Peroxisome proliferator-activated receptor gamma coactivator 1-alpha (PGC-1α)	Decrease in mitochondrial biogenesis	
	Dimethylarsinous acid (DMAIII), and	Mitochondrial transcription factor A (TFAM)		
	Dimethylarsinic acid (DMAV)	Pyruvate dehydrogenase		
	Arsenic triglutathione			

**Table 3 antioxidants-11-00071-t003:** Gut microbial interactions with metals, both antagonistic and synergistic.

Probiotic Microorganism	Toxic Element	Model/Cell Line	Treatment Effects	Citation
*Lactobacillus rhamnosus GR-1*	Arsenic, Cadmium, Mercury, Lead	Humans	Reduction in toxic levels in blood and microbiota changes (Succinivibrionaceae and Gammaproteobacteria families)	[372]
Mix of microorganisms: *L. plantarum DSM 24730*, *L. acidophilus DSM 24735*, *L. paracasei DSM 24733*, *B. infantis DSM 24737*, *B. longum DSM 24736*, *L. delbrueckii subsp. bulgaricus DSM 24734*, *B. breve DSM 24732* and *Streptococcus thermophilus DSM 24731*.	Arsenic, Cadmium, Mercury, Lead	Humans	Concentration of Cd, Hg, and Pb in breast milk, lower concentration of Cd in stools from newborns treated with the probiotics	[373]
*Lactobacillus rhamnosus GR-1*	Cadmium, Lead	Caco-2 cells	Reduce translocation of toxics	[24]
*Escherichia coli Nissle 1917* (EcN-20) and (EcN-21)	Cadmium, Mercury	Rats	Increasing GSH levels, SOD activity and catalase, decrease lipid peroxidation in liver and kidney and decrease ALT, AST and ALP activities, bilirubin, creatinine and urea levels in serum	[374]
*Streptococcus thermophilus*, *Lactobacillus acidophilus* and *Bifidobacterium bifidum*	Arsenic	Rats	Ameliorating the toxic effects on testis and blood metabolites, increase antioxidant enzyme activity (glutathione-s-transferase)	[375]
*Lactobacillus kefir CIDCA 8348* and *JCM 5818*	Cadmium	Human hepatoma cell line	Increase cell viability by binding to the toxic metal	[376]
*Lactobacillus rhamnosus Rosell-11*, *Lactobacillus acidophilus Rosell-52* and *Bifidobacterium longum Rosell-175*	Cadmium	Rat hepatocytes, Rats	Decrease genotoxicity through binding to toxic metal	[377]
*Lactobacillus rhamnosus Rosell-11*, *Lactobacillus acidophilus Rosell-52* and *Bifidobacterium longum Rosell-175*	Cadmium	Rats	Increase in toxic metal secretion by feces and decrease in concentration in the blood. Normalized ALT and AST activities	[378]
*Lactobacillus plantarum L67*	Cadmium	Murine RAW 264.7 cells	Inhibits cytotoxicity and intracellular Ca2+ mobilization; suppressed the expression of AP-1 and MAPK protecting against inflammation	[379]
*Lactobacillus plantarum CCFM8610*	Cadmium	Human intestinal cell line HT-29 and mice	Protection against cell damage, reversed the disruption of tight junctions, protected against inflammation, and decreased the intestinal permeability	[380]
*Saccharomyces cerevisiae*	Cadmium	Mice	Protected against genotoxic and spermatotoxic effects	[381]
*Lactobacillus plantarum and Bacillus coagulans*	Cadmium	Rats	Decreased AST, ALT, BUN, bilirubin (increased by toxic exposition) and metal accumulation in the liver and kidney	[382]
*Lactobacillus delbrueckii*, *Lactobacillus fermentum*, *Lactobacillus acidophilus*, *Bifidobacterium* and *Lactobacillus bulgaricus*	Cadmium	Mice	Increased toxic excretion in feces, and increasing β-catenin and BDNF in brain tissue	[383]
*Streptococcus thermophilus*	Cadmium	Mice	Through toxic metal binding, decreased levels of toxic metal in blood and attenuation levels of MDA and GSH	[384]
*Pediococcus pentosaceus GS4*	Cadmium	Mice	Reduced tissue deposition, increased fecal secretion of toxic, and increased enzymatic activity	[208]
*Bacillus cereus*	Cadmium	Fish	Reduced Cd accumulation in organs, modulate antioxidant activity and intestinal microbial composition	[385]
Lactobacillus plantarum *CCFM8610*	Cadmium	HT-29 cells, mice	Inhibition of metal absorption, alleviated cytotoxicity, reversal of the disruption of tight junctions and inhibition of inflammation	[380]
*Lactobacillus plantarum CCFM8610*	Cadmium	Fish	Mitigation of oxidative stress in tissues and reversed alterations in hematological and biochemical parameters	[386]
*Lactobacillus acidophilus*	Cadmium	Fish	Decreased number of micronucleus formation in erythrocytes and improvement of animal survival rate.	[387]
*Akkermansia muciniphila*	Cadmium	Mice	Modulation of gut microbiota composition	[388]
*Lactobacillus plantarum CCFM8661*	Lead	Mice	Decreased toxic levels in blood and tissues, recover blood δ-aminolevulinic acid dehydratase activity, avoidance of alterations in glutathione, glutathione peroxidase, superoxide dismutase, malondialdehyde, and reactive oxygen species	[389]
*Lactobacillus plantarum CCFM8662*	Lead	Mice	Induced fecal metal excretion through hepatic bile acids synthesis, enhanced biliary glutathione and increased fecal bile acid excretion.	[390]
*Faecalibacterium prausnitzii* and *Oscillibacter ruminantium*	Lead	Mice	Increase in the expression of TJ proteins (ZO-1, occludin and claudin-1), increased fecal Pb excretion, increase SCFAs, pH and oxidative reduction in the intestinal lumen	[391]
*Lactobacillus plantarum CCFM8610*	Cadmium	Mice	Decrease intestinal metal absorption, tissue accumulation, oxidative stress and ameliorate hepatic histopathological changes	[392]
*Lactobacillus reuteri*	Lead	Fish	Decreased oxidative stress, reversed alterations in hemato-biochemical parameters, and restored intestinal enzymatic activities	[393]
*Lactobacillus plantarum*, *Lactobacillus acidophilus*, *Bacillus subtilis*, *Pediococcus pentosaceus*, *Bacillus licheniformis* and *Saccharomyces cerevisiae*	Lead	Broiler Chicks	Improved antioxidant parameters, liver transaminases, decreased accumulation of metals and morphological alterations	[394]
*Lactobacillus delbrueckii subsp. bulgaricus KLDS1.0207*	Lead	Mice	Increased fecal excretion, decreased oxidative stress, lipid peroxide by decreasing MDA concentration, and improved antioxidant production	[395]
*Lactobacillus pentosus ITA23* and *Lactobacillus acidipiscis ITA44*	Lead	Broiler Chicks	Reduced metal tissue accumulation, decreased lipid peroxidation, and normalized antioxidant activity	[179]
*Lactobacillus brevis 23017*	Mercury	Mice	Reduced intestinal inflammation and decreased oxidative stress	[396]
and *Lactobacillus plantarum*	Mercury	Rats	Decreased tissue accumulation of toxic metal, avoided antioxidant alterations, normalized creatinine, bilirubin, urea AST, and ALT levels	[397]
*Streptococcus thermophilus*, *Lactobacillus acidophilus* and *Bifidobacterium bifidum*	Mercury	Rats	Protection against the adverse effects in the brain and kidney. Increased activity of glutathione-S-transferase, lactate dehydrogenase, normalized creatinine, triglycerides levels and modulate histopathological changes in the brain.	[398]

**Table 4 antioxidants-11-00071-t004:** Interactions of probiotic microbiota and their effects on the gut–brain axis.

Probiotic Microorganism	Model	Outcome	Citation
*Lactobacillus casei Shirota*	Humans	Improves stool consistency and bowel habits	[407]
*Lactobacillus acidophilus*, *Lactobacillus reuteri*, *Lactobacillus fermentum* and *Bifidobacterium bifidum*	Humans	Decreases The Movement Disorders Society-Unified Parkinson’s Disease Rating Scale (MDS-UPDRS), biomarkers of inflammation and oxidative stress (high-sensitivity C-reactive protein (hs-CRP), Malondialdehyde (MDA), Glutathione) and insulin metabolism.	[408]
*Lactobacillus acidophilus*, *Bifidobacterium bifidum*, *L. reuteri* and *Lactobacillus fermentum*	Humans	Downregulates gene expression levels of IL-1, IL-8 and TNF-α; increases the gene expression of TGF-β and PPAR-γ	[409]
*Lactobacillus acidophilus* and *Bifidobacterium infantis*	Humans	Improves abdominal pain, bloating and constipation with incomplete evacuation	[410]
*Streptococcus salivarius subsp thermophilus*, *Enterococcus faecium, Lactobacillus rhamnosus GG, Lactobacillus acidophilus, Lactobacillus plantarum, Lactobacillus paracasei, Lactobacillus delbrueckii subsp bulgaricus, and Bifidobacterium (breve* and *animalis subsp lactis*)	Humans	Improves constipation in PD patients	[411]
Lactobacillus salivarius LS01 and *Lactobacillus acidophilus*	Peripheral blood mononuclear cells from PD patients and Caco-2 cells	Reduces proinflammatory cytokines, oxidative stress and increased the anti-inflammatory response and protection of the epithelium from gut permeability	[412]
*Lactobacillus acidophilus*, *Lactobacillus rhamnosus GG* and *Bifidobacterium animalis lactis*	Mice	Increases levels of BDNF, GDNF and dopamine, and decreases levels of MAO-B in the brain.	[413]
*Bifidobacterium bifidum*, *Bifidobacterium longum*, *Lactobacillus rhamnosus*, *Lactobacillus rhamnosus GG*, *Lactobacillus plantarum LP28* and *Lactococcus lactis subsp. Lactis*	Mice	Preserves dopamine neurons, reduces the motor impairments, and maintains tyrosine hydroxylase in neurons	[414]

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
