# Peer review of "Parkinson’s Disease and the Metal–Microbiome–Gut–Brain Axis: A Systems Toxicology Approach"

_antioxidants, 2021, doi:10.3390/antiox11010071_

Round 1

Reviewer 1 Report

This is a very interesting review with the title “Parkinson’s Disease and the Metal-Microbiome-Gut-Brain Axis: A Systems Toxicology Approach”. It is a large and complete review based on a high number of references. Nevertheless, I have some concerns:

-There is a lack of a general overview of this field regarding recent research studies in PD human samples.

-The present manuscript cites a high proportion of references that consist of reviews.

-In addition, these references focused on a large variability of pathologies, samples, patients, animal models…

-In general, few original research articles based on experimental research about the relationship between PD and metal-microbiome are discussed.

-Moreover, only a few of them are more or less recently published (from 2016 to 2021).

Regarding to these points, I would suggest to carry out a specific search with defined keywords according to the manuscript title, excluding reviews, and including only research papers between 2015 and 2021. In this sense, the authors could include a sub-section discussing recent research in this specific field.

Reviewer 2 Report

This is a thorough presentation of research relating to the regulation of metals via the gut-brain-axis and its implications for PD risk.  The role of the microbiome is an exciting growing area of understanding of neurologic (and other) disorders.  The review appears timely.  If other similar reviews have appeared, the reviewer is not aware of them.

This is ambitious work that also has an aspect of meta-analysis.  By reviewing a larger body of work on PD, metals, and the gut brain axis, the authors are working to support the hypothesis that poor metal clearing in the microbiome can be linked to PD.

My main area of feedback is that to me, there was an identity crisis in the paper where at times it looked more like a review, and at other times, it read more like a meta-analysis to support the hypothesis about metals/Gut-Brain-Axis/PD.  Certainly I agree that it looks more like a review than a meta-analysis, but I advise that the authors should do some wordsmithing to clarify a bit more directly the purpose of this work: to be more of more of a review or more of a meta-analysis?  It can have aspects of both, but I think this needs to be addressed a more.  

Comment: Speaking briefly to omics, it seems that BCAAs continue to be difficult to interpret outside of metabolic syndrome/diabetes.  There may be a connection between obesity and neurologic disorders like PD, but it has been challenging to resolve that question.  But it leaves open the hypothesis that changing BCAAs may report more on T2D risk and progression rather than directly to PD.

The manuscript is well written particularly in expressing complex findings concisely and clearly while taking in a large body of work (>400 citations).  I think the explanations/summaries of the cited works are well crafted.  But there were many typos and small sentence errors present throughout.  This is understandable for such a large manuscript.  But it is vital that the authors make a thorough editing.  The reviewer started to list them, but felt unable to take this on for such a large manuscript.

Please check if all acronyms are defined on first appearance.

Minor:

Correction: delete abnormal in ‘Abnormal protein misfolding’

I found the layout of Table 1 confusing.  

Underlining in Table 3 very distracting.

I know this will seem nitpicky, but the phrases ‘play a role’ or ‘play roles’ have become overused in science.  Suggest to reword, “neuron enteric cells play essential roles in gastrointestinal motility” to “ neuron enteric cells are essential to gastrointestinal motility”.

Overall, I often forgot I was reviewing this while reading it as I found it interesting.  This appears to be a strong contribution to review important recent results and support an emerging understanding of metals and gut-brain-axis in PD.

Round 2

Reviewer 1 Report

The authors have answered all the required points.